# FSAP Protects against Histone-Mediated Increase in Endothelial Permeability In Vitro

**DOI:** 10.3390/ijms232213706

**Published:** 2022-11-08

**Authors:** Xue Yan Cui, Benedicte Stavik, Bernd Thiede, Per Morten Sandset, Sandip M. Kanse

**Affiliations:** 1Department of Haematology, Oslo University Hospital, 0424 Oslo, Norway; 2Research Institute of Internal Medicine, Oslo University Hospital, 0424 Oslo, Norway; 3Institute of Clinical Medicine, University of Oslo, 0315 Oslo, Norway; 4Department of Haematology, The First Affiliated Hospital of Zhengzhou University, Zhengzhou 450052, China; 5Department of Biosciences, University of Oslo, 0315 Oslo, Norway; 6Institute of Basic Medical Sciences, University of Oslo, 0315 Oslo, Norway

**Keywords:** FSAP, endothelium, histone, permeability, TLR

## Abstract

Factor-VII-activating protease (FSAP) is involved in the regulation of hemostasis and inflammation. Extracellular histones play a role in inflammation and the conversion of latent pro-FSAP into active FSAP. FSAP has been shown to regulate endothelial permeability, but the mechanisms are not clear. Here, we have investigated the effects of FSAP on endothelial permeability in vitro. A mixture of histones from calf thymus stimulated permeability, and the wild-type (WT) serine protease domain (SPD) of FSAP blocked this effect. WT–SPD–FSAP did not influence permeability on its own, nor that stimulated by thrombin or vascular endothelial growth factor (VEGF)-A_165_. Histones induced a large-scale rearrangement of the junction proteins VE-cadherin and zona occludens-1 from a clear junctional distribution to a diffuse pattern. The presence of WT–SPD–FSAP inhibited these changes. Permeability changes by histones were blocked by both TLR-2 and TLR4 blocking antibodies. Histones upregulated the expression of TLR-2, but not TLR-4, in HUVEC cells, and WT–SPD–FSAP abolished the upregulation of TLR-2 expression. An inactive variant, Marburg I (MI)–SPD–FSAP, did not have any of these effects. The inhibition of histone-mediated permeability may be an important function of FSAP with relevance to sepsis, trauma, and stroke and the need to be investigated further in in vivo experiments.

## 1. Introduction

Endothelial cells constitute the inner lining of all blood vessels and regulate the interface between the circulating blood and the vessel wall. They directly control the vascular barrier function, the passive and active transport of blood-borne substances, fluid homeostasis, vascular smooth muscle tone, leukocyte transmigration, blood coagulation, and fibrinolysis. The disruption of this selective cellular barrier is a critical feature of inflammation and an important contributing factor to many vascular diseases as well as tumor progression [1]. Barrier properties are maintained by the polarization of endothelial cells into apical and basolateral direction as well as the expression of different types of junctional proteins and accessory cells such as pericytes and astrocytes [1].

Several inflammatory mediators affect endothelial permeability, e.g., vascular endothelial growth factor (VEGF), thrombin, histamine, bradykinin, angiopoietins, sphingosine-1-phosphate, ATP, and histones [1]. Thrombin is the key effector serine protease in blood coagulation, and it induces a rapid and reversible increase in endothelial permeability through protease activated receptors (PARs) [2]. Histones are nuclear cationic proteins (H1, H2A, H2B, H3, and H4), which, together with DNA, constitute the nucleosomes [3]. Histones are released extracellularly upon necrotic- or programmed-cell death or from activated neutrophils as part of neutrophil extracellular traps (NETosis) [4]. Extracellular histones are pro-thrombotic, toxic, and pro-inflammatory, and they increase endothelial permeability [5,6]. Histones disturb cell membrane architecture, membrane permeability, calcium influx and oxidative stress through interactions with cell membrane phospholipids as well as through toll-like receptor (TLR) 2 (TLR-2) and TLR-4 [7] expressed on endothelial cells [8].

Factor-VII-activating protease (FSAP) is a circulating serine protease involved in regulating hemostasis, inflammation, and tissue remodeling. Single nucleotide polymorphisms (SNPs) in or near the FSAP-gene locus result in an increased risk of stroke [9,10]. Of these SNP’s, the Marburg-I (MI) SNP, results in a protein with low proteolytic activity [11]. Pro-FSAP, an inactive single-chain zymogen, is converted into the two-chain active protease by auto proteolysis upon contact with extracellular histones [12], and histone toxicity is neutralized by FSAP through their proteolytic degradation [13]. Based on these results, we hypothesized that FSAP neutralizes the permeability-related effects of histones released upon tissue injury or inflammation. We explored the effect of the serine protease domain (SPD) of FSAP on histone-induced endothelial permeability in vitro. The effect of the wild-type (WT)–SPD–FSAP was compared to the MI–SPD–FSAP, and the role of TLR-2 and -4 in the effect of histones was investigated. The proteolytic degradation of histones was characterized by mass spectrometry.

## 2. Results

### 2.1. FSAP Inhibits Permeability Changes by Histones

We first tested if histones induced permeability changes and if this was modulated by FSAP. Our results show that extracellular histones increased permeability in HUVECs and HAoECs at 24 h (Figure 1A). WT–SPD–FSAP itself showed no effect on the permeability of HUVECs after 24 h treatment (Figure 1B). However, WT–SPD–FSAP abolished the histone-induced increase in permeability, while the inactive isoform MI–SPD–FSAP had no effect (Figure 1C).

### 2.2. FSAP Does Not Influence Permeability Changes by Thrombin and VEGF-A_165_

We then examined the effect of WT–SPD–FSAP on other permeability-modulating factors. The permeability of HUVECs was strongly increased by 2 h treatment of thrombin as well as the 24 h treatment of VEGF-A_165_. However, WT–SPD–FSAP did not modulate the effect of thrombin- or VEGF-induced permeability (Figure 2A,B).

### 2.3. FSAP Rescues Redistribution of Junctional Proteins by Histones

Permeability changes are often due to a disassembly of junctional protein complexes, and we examined these changes by immunofluorescence analysis. VE-cadherin (adherence junctions) and ZO-1 (tight junctions) were redistributed from the cell–cell contacts into a more diffuse distribution in the presence of histones. In the concomitant presence of WT–SPD–FSAP, histones did not alter the junctional distribution pattern of these proteins. The inactive isoform MI–SPD–FSAP did not show these effects (Figure 3).

### 2.4. Role of TLR in the Histone-Mediated Modulation of Permeability

Considering that TLR-2 and -4 are major histone receptors, we next designed experiments to test if TLR-2 and -4 are involved in the actions of histones. Both TLR2- and TLR4-blocking antibodies inhibited the permeability induced by histones (Figure 4A,B). Endothelial cells express mRNA for both of these receptors; TLR2 mRNA expression was strongly up-regulated by extracellular histones, WT–SPD–FSAP abolished this effect, and MI–SPD–FSAP had no effect (Figure 4C). No significant increase in TLR4 mRNA expression was induced by histones (Figure 4D).

### 2.5. The Cleavage of Histones by WT–SPD–FSAP

From our experiments, WT–SPD–FSAP cleaved H1, H2A/H2B, and H3 at IC_50_ of 2 ± 0.03 µg/mL, 26 ± 0.3 µg/mL, and 10 ± 0.7 µg/mL, respectively (n = 3 experiments) (Appendix A). The cleavage of H4 is not determined because it co-migrates with the degradation products of the other histones and there is an increase in band density. FSAP cleaves after basic amino acids [14]; there were multiple cleavage sites in each of the histone examined, and these were distributed evenly across the whole molecule (Appendix A).

## 3. Discussion

A multitude of studies over the last decade have shown that enhanced tissue necrosis, programmed cell death, inflammation, and NETosis can lead to the release of extracellular histones [4]. These are toxic to cells and are strongly pro-inflammatory and function as damage-associated molecular pattern proteins (DAMPs). The effects of DAMPs are counteracted, directly or indirectly, by suppressors of DAMPs (SAMPs) [15]. Histones are neutralized by proteolytic degradation, in which FSAP has a prominent effect and thus can be defined as a SAMP [12]. In this in vitro study, we demonstrate that the endothelial permeability stimulating effect of histones is inhibited by SPD–FSAP.

For these studies, we used a natural source of histones isolated from calf thymus so that the ability to form multimeric complexes would be possible, as is normal for these proteins. Furthermore, this complex contains many of the isoforms of histones as well as retaining post-translational modifications. Isolated H3 and H4 have been shown to regulate endothelial permeability but not H1, H2A, or H2B [16]. However, the propensity of histones to form complexes with one another means that they are unlikely to exist as monomers in vivo. Of the different isoforms of histones, we found that the linker histone H1 was the most effective substrate for SPD–FSAP, whereas histone H2A/H2B was less effective. It might be that the core histones are in a multimeric complex and the linker histone is not, thus making the latter more susceptible to cleavage. Previous studies have shown that FSAP is more potent than activated protein C in neutralizing the activity of histones [13,17]. The mass spectrometry showed that each histone in the mixture underwent cleavage at multiple sites. This indiscriminate cleavage of histones is distinct from the very selective cleavage of other substrates such as PARs [18], the tissue factor pathways inhibitor (TFPI) [18], the epithelial sodium channel [19], and VEGF-A_165_ [20].

An earlier study investigating the effect of FSAP on endothelial permeability used the approach of blocking endogenous FSAP expression in endothelial cells [21]. However, most publicly accessible databases and our own results indicate that endothelial cells do not express any FSAP, which makes it difficult to interpret these results. One possible explanation for these results is that the cultivation of cells in fetal calf serum may lead to the association of serum-derived FSAP with cells and these changes during siRNA transfection. The author’s conclusion that the effect of FSAP is mediated by Rho-GTPases and PARs maybe due to the fact that the inhibition of these pathways may have a general inhibitory effect on permeability independent of the stimulus used. For our studies, we used recombinant SPD–FSAP and not the full-length protein because its expression in sufficient amounts is not currently possible. It is possible that this isolated domain has different properties compared to full-length FSAP. Our previous studies show that SPD-FSAP cleaves substrates, including PARs [18], histones (this study), pro-urokinase, factor VII, the tissue-factor-pathway inhibitor [22], and the epithelial sodium channel [19] in a similar way to plasma-purified FSAP.

FSAP has been shown to modulate hyaluronic acid-mediated alterations in endothelial permeability via PAR-1 and PAR-3 [21]. Thrombin- or VEGF-A_165_- induced permeability was not influenced by SPD–FSAP. VEGF-A_165_ is reported to be cleaved by FSAP, altering its in vivo properties related to angiogenesis but not its in vitro properties on endothelial cells with respect to proliferation and migration [20]. The results on endothelial permeability are in line with these observations.

Protein complexes in the tight junction and adherens junction regulate the permeability of the endothelium and are under the control of signal transduction pathways, including the Rho GTPases [23]. The junctional distribution pattern of these proteins, VE-cadherin and ZO-1, was disturbed by histones. Morphological analysis indicates that this is most likely due to a combination of effects on the disassembly of cellular junction proteins and a general alteration in cell morphology due to histone toxicity and signaling via TLR-2 and TLR-4. It has been demonstrated that the stimulation of endothelial cells with histones also leads to changes in the expression of cytokines and inflammatory molecules [24]. Interestingly, histones upregulated the mRNA expression of TLR-2 but not TLR-4. The histone-mediated regulation of (i) permeability, (ii) the distribution of junctional proteins, and (iii) TLR-2mRNA expression were reversed by SPD–FSAP. These results are is in line with our previous conclusions that FSAP also reverses histone cytotoxicity on endothelial cells [17].

FSAP-deficient mice subjected to thromboembolic stroke exhibit a larger infarct volume [25], which indicates that FSAP has a protective role of FSAP in the stroke, and this was confirmed with the use of exogenous FSAP [26]. In this model, the decrease in infarct area coincided with a decrease in the area of infarct hemisphere. This change in hemisphere size is due to edema and would suggest that SPD–FSAP inhibits permeability in vivo. Further in vivo studies that specifically measure vasogenic edema in the presence of endogenously released histones and FSAP are required to consolidate these findings.

## 4. Materials and Methods

Materials: Recombinant WT–SPD–FSAP and MI–SPD–FSAP were prepared as described before [22]. Histones (H9250) and thrombin from human plasma (T6884) were purchased from Sigma Aldrich (St. Louis, MO, USA). VEGF-A_16_ was from R&D systems Inc. (293-VE-010, Minneapolies, MN, USA).

Cell culture: Human umbilical vein endothelial cells (HUVECs, CC-2517, Lonza, Basel, Switzerland) and human aortic endothelial cells (HAoECs, Promocell, Heidelberg, Germany) were grown in endothelial cell growth medium MV2 with supplements (c-22011, Promocell). Cells were passaged with detach kit 2 (Promocell), and only low passage number cells were used.

Macromolecular permeability assays: These were performed in transwell insert plates (24-well, 6.5 mm, 0.4 µm pore size; Costar (Corning, Kennebunk, ME, USA). 2 × 10^5^ HUVECs or HAoECs were seeded out in 300 µL growth medium on the transwells, and 1 mL growth medium was added in the lower chamber. Cells were grown for 4 days until they were confluent. Then, the cells were washed once with basal medium MV2 (c-22211, Promocell) before they were treated with VEGF-A_165_ or histones and/or SPD–FSAP for 16 h. Experiments with thrombin were done for 2 h. Permeability was measured by adding streptavidin-horseradish peroxidase (HRP) (DY998, dilution 1:200, R&D Systems Inc., Minneapolis, MN, USA) onto the endothelial monolayer and incubated for 10 min at 37 °C. Substrate 3, 3′, 5, 5′-tetramethylbenzidine (TMB, T0440, Sigma Aldrich, St. Louis, MO, USA) was added, and the absorption at 450 nm was read after adding 2 N H_2_SO_4_ water solution to stop the reaction [27]. For blocking experiments, anti-TLR-2 antibody (TL2.1) was from Invitrogen (14-9922-82, Carlsbad, CA, USA) and anti-TLR-4 antibody (HTA125) was from Abcam (ab30667, Cambridge, UK). Mouse IgG2a (MAB003) was from R&D Systems Inc. (Minneapolis, MN, USA).

RNA isolation, cDNA synthesis, and relative quantification of mRNA: Total RNA was extracted, quantified, and reverse-transcribed to cDNA, as described before [28]. Quantitative RT-PCR was used to measure the relative mRNA expression of TLR-2 (Hs00152932_m1) and -4 (Hs00152939_m1) (Applied Biosystems, Waltham, MA, USA). Ct values were normalized against the endogenous control *TBP* (4325803, Applied Biosystems). Negative controls without cDNA were always included.

Immunofluorescence staining: HUVECs were subjected to different treatments, as described in figure legend, and then fixed by methanol for 10 min at 4 °C. Vascular endothelial cadherin (VE-cadherin) and zonula occludens-1 (ZO-1) were used to determine the integrity of intercellular junctions. The fixed cells were incubated with anti-VE-cadherin (MAB9381, R&D Systems Inc., Minneapolis, MN, USA) or anti-ZO-1 (40-2200, Invitrogen, Thermo Fisher Scientific, Rockford, IL, USA) primary antibodies overnight at 4 °C. Normal rabbit IgG (#2729, Cell Signaling Technology, Danvers, MA, USA) and mouse IgG1 (MAB002, R&D Systems, Minneapolis, MN, USA) were used as controls. Following washing with Dulbecco’s phosphate buffered saline (Sigma Aldrich) in 2% fetal calf serum, cells were stained with Alexa Fluor 488-labeled donkey anti-rabbit (A21206) and Alexa Fluor 568-labeled donkey anti-mouse (A10037) secondary antibodies (Thermo Fisher Scientific, Eugene, OR, USA), for 90 min at room temperature. Nuclei were stained with diamidino-2-phenylindole (ProLong Gold antifade reagent, Thermo Fisher Scientific). Fluorescence images were acquired by using a Leica microscope. Magnification was 40×.

Statistical analysis: Each experiment was performed in triplicate, and the results are shown as mean ± SEM. For the permeability experiments, the results from three independent experiments are pooled and are shown as the mean ± SEM, and the statistical analysis was carried out using one-way analysis of variance (ANOVA) followed by Bonferroni post test using Graphpad Prism.

## Figures and Tables

**Figure 1 ijms-23-13706-f001:**
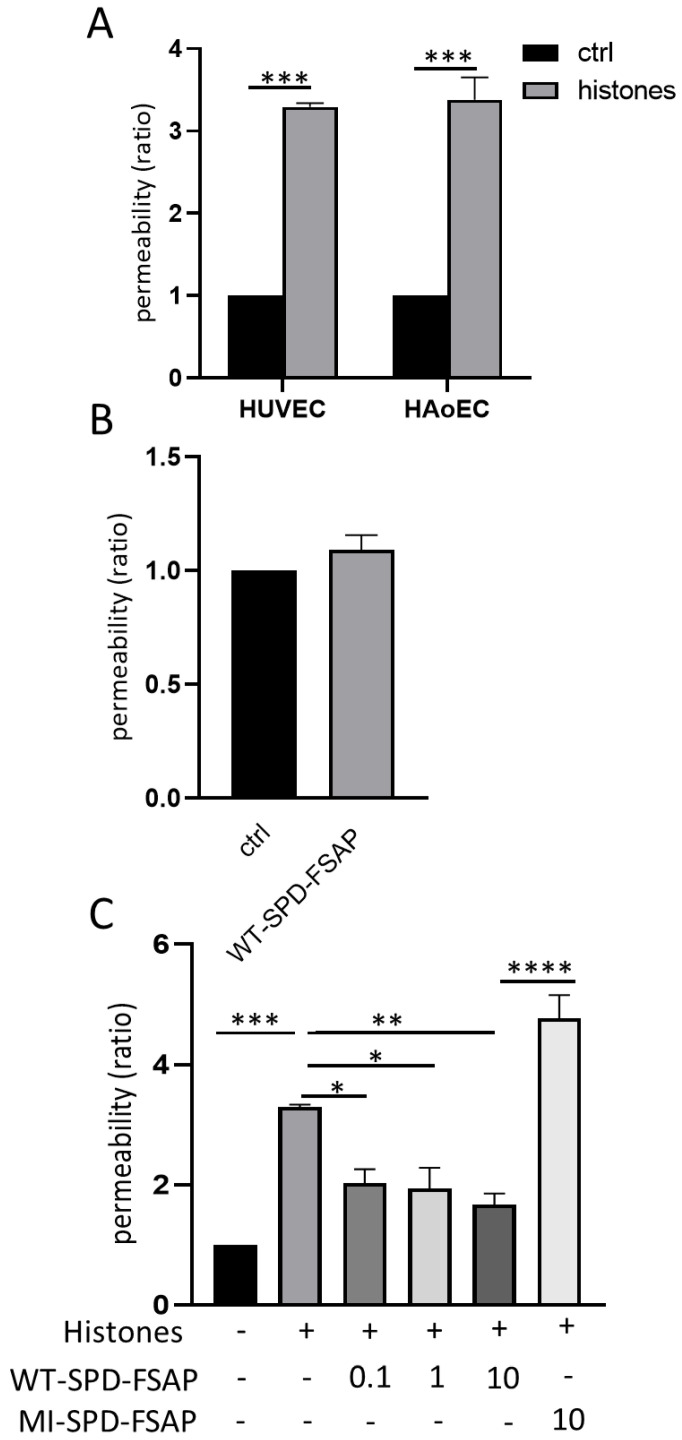
Effect of SPD-FSAP on histones-induced permeability. (**A**) Histones (50 µg/mL) induced the permeability of both HUVECs and HAoECs. Permeability ratio was calculated as the rate relative to control. Data are presented as mean ± SEM; n = 3, *** *p* < 0.001. (**B**) WT–SPD–FSAP (10 µg/mL) had no effect on the permeability of HUVECs. (**C**) 0.1, 1, and 10 µg/mL WT–SPD–FSAP repressed the increased permeability induced by 50 µg/mL histones, while 10 µg/mL MI–SPD–FSAP had no effect. Data are presented as mean ± SEM; n = 3, * *p* < 0.05, ** *p* < 0.01, *** *p* < 0.001, and **** *p* < 0.0001. N represents 3 independent experiments, with 3 replicates in each experiment.

**Figure 2 ijms-23-13706-f002:**
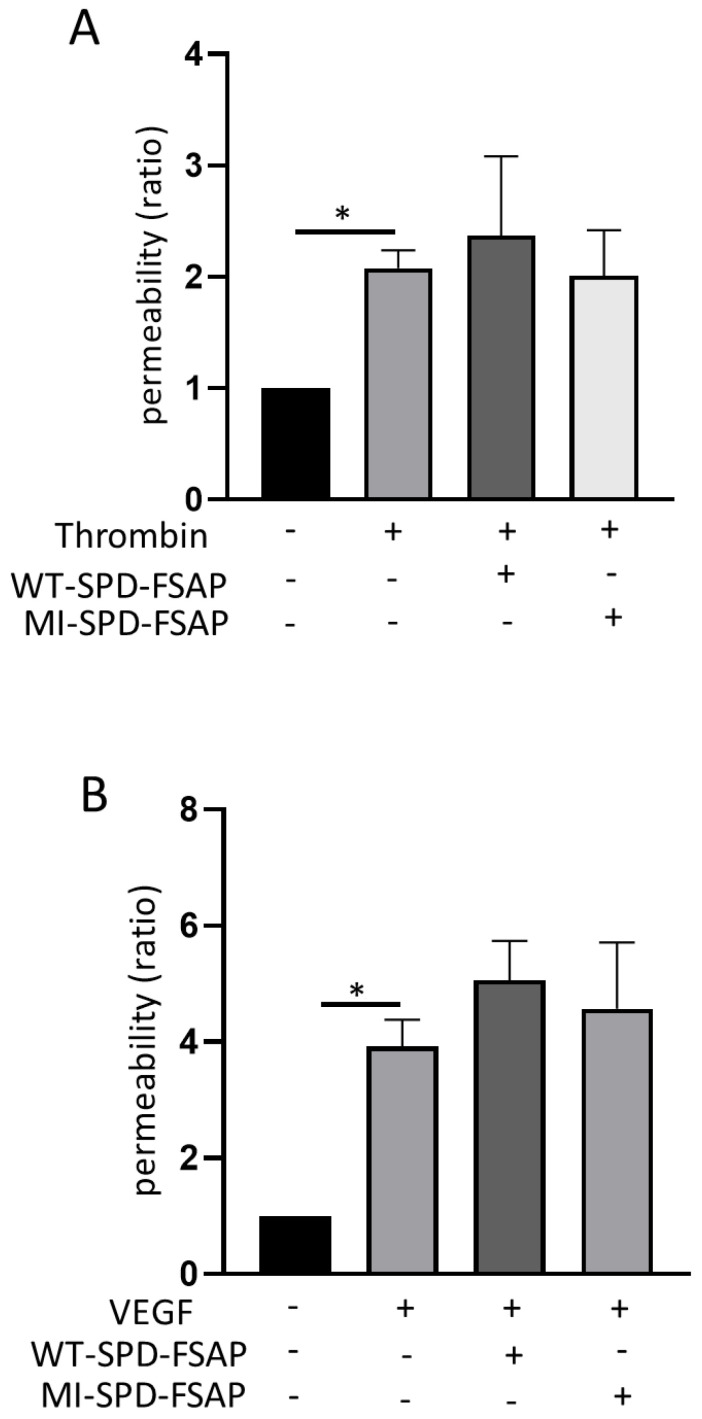
Effect of SPD–FSAP on thrombin- or VEGF-induced permeability. (**A**) 2 unit/mL thrombin with/without 10 µg/mL recombinant WT–SPD–FSAP or MI–SPD–FSAP were applied to HUVECs for 2 h. (**B**) 50 ng/mL VEGF with/without 10 µg/mL recombinant WT–SPD–FSAP or MI–SPD–FSAP were applied to HUVECs for 16 h. Permeability ratio was calculated as the rate relative to control. Data are presented as mean ± SEM; n = 3. * *p* < 0.05 N represents 3 independent experiments, with 3 replicates in each experiment.

**Figure 3 ijms-23-13706-f003:**
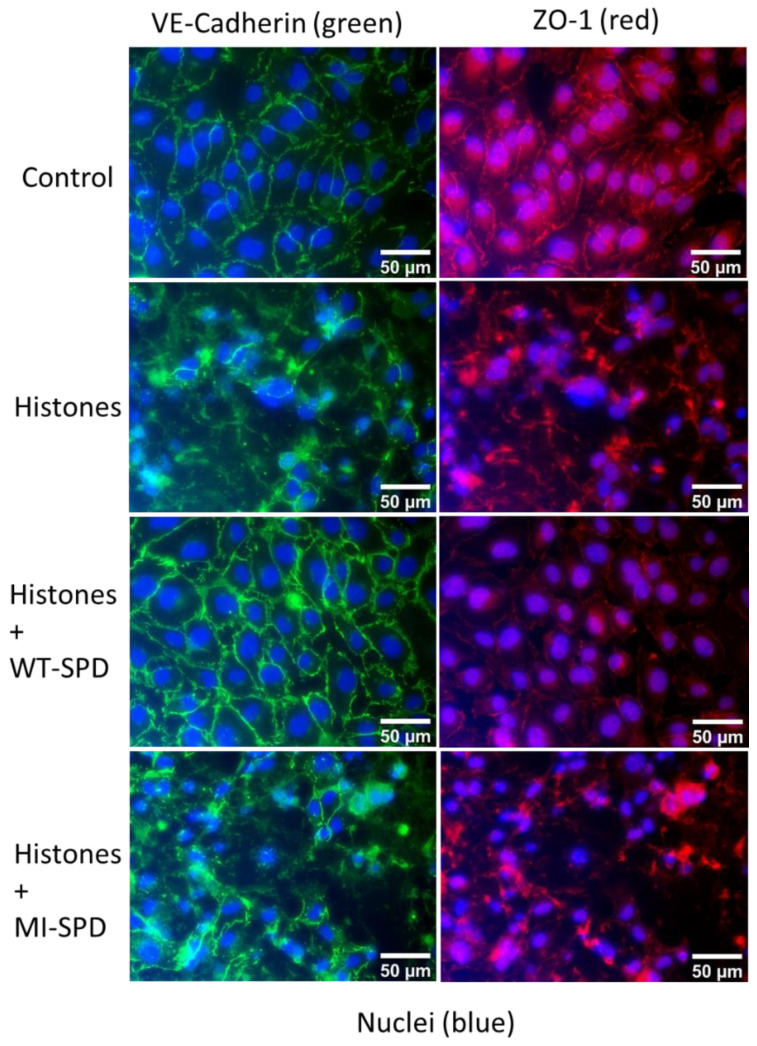
Effect of SPD–FSAP on the pattern of junctional proteins. HUVEC were treated with 50 μg/mL histones and/or 10 μg/mL WT–SPD–FSAP for 24 h before fixation. MI–SPD–FSAP was used as a negative control. Immunofluorescent staining of the junction protein VE-cadherin (green), ZO-1 (red), and nuclei (blue). Relative IgG was used as control. Representative images from 3 independent experiments.

**Figure 4 ijms-23-13706-f004:**
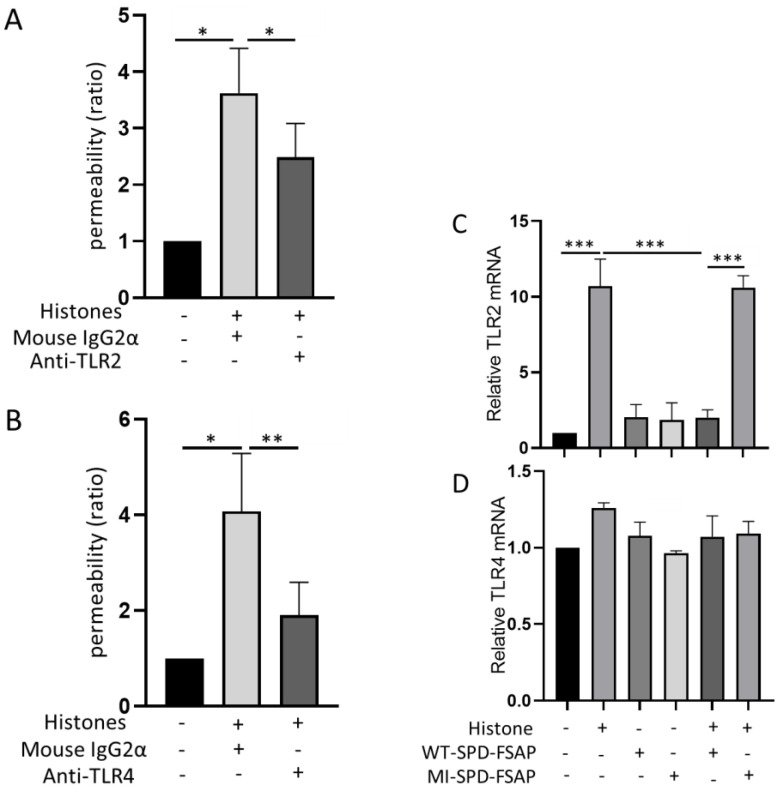
Role of SPD-FSAP in the overexpression of TLR4 induced by histones. (**A**,**B**) Permeability induced by histones was decreased significantly by antibodies against both TLR2 and TLR4. (**C**) Quantitative RT-PCR showed that TLR2 mRNA expression was strongly increased by histones. WT–SPD–FSAP abolished this effect, while MI–SPD–FSAP did not. (**D**) TLR2 mRNA expression was not influenced by histones and/or WT–SPD–FSAP. Data are presented as mean ± SEM; n = 3, * *p* < 0.05, ** *p* < 0.01, and *** *p* < 0.001. N represents 3 independent experiments, with 3 replicates in each experiment.

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
