# Peer review of "FSAP Protects against Histone-Mediated Increase in Endothelial Permeability In Vitro"

_ijms, 2022, doi:10.3390/ijms232213706_

Round 1
Reviewer 1 Report
The article by Kanse group reports the effect of FSAP against histone-mediated increase in endothelial permeability in vitro. The work is interesting, and the conclusion is well supported by the results. I have few suggestions and comments to make the manuscript more understandable to a broader scientific community.
1. Describe in brief if there is any previous reports on similar work.
2. Describe the function of the proteins/protein complexes/antibodies (eg. HUVECs, HAoECs, VE-cadherin, ZO-1) in brief.
3. Avoid using jargons.
4. Make sure you use the full form of abbreviations once when you use the term.
Specific comments:
1. Explain what is MI-SPD-FSAP.
2. Why did you focus on HUVAC cells for the effect of SPD-FSAPs?
3. Mention about controls (eg. IgG) in brief in the results sections.
4. In figure S1 why H4 is missing in the right panel? Why H4 is not mentioned in the caption for left figure?
5. Line 55, insert space between regulating and hemostasis.
6. Few words are having bigger font size than the rest. Please check.
Author Response
Response to Reviewer 1 Comments
Point 1: Describe in brief if there is any previous reports on similar work.
Response 1: FSAP has been shown to modulate hyaluronic acid-mediated alterations in endothelial permeability [1]. This study investigates the effect of FSAP on endothelial permeability by using the approach of blocking endogenous FSAP expression in endothelial cells. However, most publicly accessible databases and our own results indicate that endothelial cells do not express any FSAP, which makes it difficult to interpret their results. One possible explanation for their results could be that cultivation of cells in fetal calf serum may lead to the association of serum-derived FSAP with cells which changes after siRNA transfection. Their conclusion that the effect of FSAP is mediated by Rho-GTPases and PARs maybe due to the fact that inhibition of these pathways may have a general inhibitory effect on permeability independent of the stimulus used. For our studies, we used recombinant SPD domain of FSAP and not the full-length protein because its expression in sufficient amounts is not currently possible. That this isolated domain may have different properties compared to full length FSAP is possible. Our previous studies show that all the substrates we have tested including PARs [2], histones (this study), pro-urokinase, Factor VII and Tissue factor pathway inhibitor [3], epithelial sodium channel [4] are cleaved by SPD in a similar way to plasma-purified FSAP. This is described in the discussion part (page 7, line 157-165). Besides this, the authors are not aware of any other reports on FSAP in relation to endothelial permeability.
Point 2: Describe the function of the proteins/protein complexes/antibodies (eg. HUVECs, HAoECs, VE-cadherin, ZO-1) in brief.
Response 2: HUVECs are human umbilical vein endothelial cells and HAoECs are human aortic endothelial cells (Lonza), and they are described in the Materials and Methods part (page 8, line 202-204). These cell lines were used for endothelial permeability assays.
VE-cadherin is a vascular endothelial cadherin, also known as Cadherin 5, type 2 or CD144. VE-cadherin-based adherens junctions are particularly important for the integrity of intercellular junctions that determines the permeability of the endothelium. ZO-1 is zonula occludens-1, also known as tight junction protein-1 and it is one of the tight junction-associated proteins. VE-cadherin and zo-1 were used to visualize the junctions of endothelial cells. We have added these details in the manuscript (page 9, line 234-236)
Point 3: Avoid using jargons.
Response 3: Thank you for correcting the jargons in the manuscript. No jargon is used in the manuscript now.
Point 4: Make sure you use the full form of abbreviations once when you use the term.
Response 4: We have corrected them in the manuscript (page 2, line 72; page 3, line 77 and 79; page 4, line 91 and 93; page 5, line 104, 107 and 114; page 7, line 119, 142,150 and 167; page 8, line 172, 175, 189, 196, 205, 210, 211 and 214; page 9, line 235)
Specific comments:
Point 1: Explain what is MI-SPD-FSAP.
Response 1: One of the single nucleotide polymorphisms (SNPs) in the serine protease domain (SPD) of FSAP gene locus is called Marburg-I (MI), and results in a protein with low proteolytic activity [5]. Therefore, MI-SPD-FSAP is an inactive form of FSAP peptide and used as a negative control in our study. We had described it in the introduction part (page 2, line 56-58).
Point 2: Why did you focus on HUVAC cells for the effect of SPD-FSAPs?
Response 2: Our previous study showed that SPD-FSAP reduces edema in the ischemic stroke area of mouse models [6], which, we consider, is due to the decreased endothelial permeability of blood vessels, especially veins or capillaries. HUVEC cells are primary vein endothelial cells isolated from the human umbilical cord and is a well-established model system for studying endothelial cell function, including angiogenesis [7], hypoxia [8] and inflammation [8, 9]. Therefore, HUVEC cells are a good model for our in vitro permeability study.
Point 3: Mention about controls (eg. IgG) in brief in the results sections.
Response 3: Thanks for comments. We have added it in the manuscript (page 5, line 107)
Point 4: In figure S1 why H4 is missing in the right panel? Why H4 is not mentioned in the caption for left figure?
Response 4: Essentially, the cleavage of other histones leads to lower MW cleavage products that co-migrate with H4. This make the quantification of H4 impossible. That is why we have excluded H4 from these analysis. We have described it on Page 7, lines 129-130.
Point 5: Line 55, insert space between regulating and hemostasis.
Response 5: Thank you for the comment. We have corrected it (page2, line 55).
Point 6: Few words are having bigger font size than the rest. Please check.
Response 6: Thank you for comments. We have correct it (page 8, line 200-203)
References
- Mambetsariev N, Mirzapoiazova T, Mambetsariev B, Sammani S, Lennon FE, Garcia JG, Singleton PA. Hyaluronic Acid binding protein 2 is a novel regulator of vascular integrity. Arterioscler Thromb Vasc Biol. 2010;30(3):483-90.
- Byskov K, Le Gall SM, Thiede B, Camerer E, Kanse SM. Protease activated receptors (PAR)-1 and -2 mediate cellular effects of factor VII activating protease (FSAP). FASEB J. 2020;34(1):1079-90.
- Nielsen NV, Roedel E, Manna D, Etscheid M, Morth JP, Kanse SM. Characterization of the enzymatic activity of the serine protease domain of Factor VII activating protease (FSAP). Sci Rep. 2019;9(1):18990.
- Artunc F, Bohnert BN, Schneider JC, Staudner T, Sure F, Ilyaskin AV, Worn M, Essigke D, Janessa A, Nielsen NV, Birkenfeld AL, Etscheid M, Haerteis S, Korbmacher C, Kanse SM. Proteolytic activation of the epithelial sodium channel (ENaC) by factor VII activating protease (FSAP) and its relevance for sodium retention in nephrotic mice. Pflugers Arch. 2022;474(2):217-29.
- Willeit J, Kiechl S, Weimer T, Mair A, Santer P, Wiedermann CJ, Roemisch J. Marburg I polymorphism of factor VII--activating protease: a prominent risk predictor of carotid stenosis. Circulation. 2003;107(5):667-70.
- Kim J, Yeon, Manna D, Etscheid M, Leergaard T, B, Kanse S, M. Factor VII activating protease (FSAP) inhibits the outcome of ischemic stroke in mouse models. FASEB J. 2022(36:e22564).
- Chrifi I, Louzao-Martinez L, Brandt MM, van Dijk CGM, Burgisser PE, Zhu C, Kros JM, Verhaar MC, Duncker DJ, Cheng C. CMTM4 regulates angiogenesis by promoting cell surface recycling of VE-cadherin to endothelial adherens junctions. Angiogenesis. 2019;22(1):75-93.
- Stavik B, Espada S, Cui XY, Iversen N, Holm S, Mowinkel MC, Halvorsen B, Skretting G, Sandset PM. EPAS1/HIF-2 alpha-mediated downregulation of tissue factor pathway inhibitor leads to a pro-thrombotic potential in endothelial cells. Biochim Biophys Acta. 2016;1862(4):670-8.
- Chen T, Zhang X, Zhu G, Liu H, Chen J, Wang Y, He X. Quercetin inhibits TNF-alpha induced HUVECs apoptosis and inflammation via downregulating NF-kB and AP-1 signaling pathway in vitro. Medicine (Baltimore). 2020;99(38):e22241.

Reviewer 2 Report
Xue Yan Cui and colleagues investigated and showed how FSAP protects against histone-mediated increase in endothelial permeability in vitro. The findings are novel and interesting. The manuscript reads well. To improve the quality of the manuscript I have only a few minor comments. Authors should measure the fluorescence intensity for the imaging data. For the methods section, please provide the catalog and lot numbers for the reagents, and kits if available.
